# Current and Emerging Methods for the Synthesis of Single-Stranded DNA

**DOI:** 10.3390/genes11020116

**Published:** 2020-01-21

**Authors:** Min Hao, Jianjun Qiao, Hao Qi

**Affiliations:** 1School of Chemical Engineering and Technology, Tianjin University, Tianjin 300072, China; min1213@tju.edu.cn (M.H.); jianjunq@tju.edu.cn (J.Q.); 2Key Laboratory of Systems Bioengineering of Ministry of Education, Tianjin University, Tianjin 300072, China; 3SynBio Research Platform, Collaborative Innovation Center of Chemical Science and Engineering, Tianjin University, Tianjin 300072, China

**Keywords:** ssDNA, chemical synthesis, enzyme synthesis, bacterial-based synthesis

## Abstract

Methods for synthesizing arbitrary single-strand DNA (ssDNA) fragments are rapidly becoming fundamental tools for gene editing, DNA origami, DNA storage, and other applications. To meet the rising application requirements, numerous methods have been developed to produce ssDNA. Some approaches allow the synthesis of freely chosen user-defined ssDNA sequences to overcome the restrictions and limitations of different length, purity, and yield. In this perspective, we provide an overview of the representative ssDNA production strategies and their most significant challenges to enable the readers to make informed choices of synthesis methods and enhance the availability of increasingly inexpensive synthetic ssDNA. We also aim to stimulate a broader interest in the continued development of efficient ssDNA synthesis techniques and improve their applications in future research.

## 1. Introduction

DNA is the carrier of genetic information and as such it is an indispensable part of basic biological research, biomaterial science, and synthetic biology. The vast majority of modern biological research and bioengineering relies on synthetic custom DNA sequences, including oligonucleotides and longer constructs, such as synthetic genes and even entire chromosomes [1,2]. Breakthroughs that enable the large-scale, low-cost, and high-efficiency construction of desired DNA sequences could catalyze rapid progress in biological research and application [3]. Today, the complete reconstruction of viral and bacterial genomes is the proof of our synthetic ability [4,5,6]. There is no doubt that user-defined DNA synthesis has improved our ability to understand the interactions between DNA and protein [7,8,9,10], uncover the structural effects of regulatory elements that drive expression [11,12], as well as engineer the structural and functional characteristic of mammalian, yeast, and bacterial systems [13,14,15,16].

Although there are scalable methods for the production of double-stranded DNA (dsDNA) both in vitro and in vivo, equally efficient methodologies for the synthesis of single-stranded DNA (ssDNA) would be desirable for a number of special applications. In fact, the synthesis of ssDNA has become an enabling technology for modern DNA-based biomaterials. In this context, ssDNA is considered an elemental material with an intriguing application potential in many biological reactions and the broad applicability of related DNA nanotechnology [17]. For instance, ssDNA can be used as the scaffold for DNA nanotechnology [18,19,20], the donor DNA for clustered regularly interspaced short palindromic repeats (CRISPR) and CRISPR-associated protein 9 (Cas9) systems CRISPR-Cas9 systems [21,22], carrier for drug delivery [23], in molecular diagnostic [24], DNA-based data storage [25,26,27], and various nanoscale applications [28]. To meet these different application requirements, a variety of methods have been developed based on different principles. Thus, it is of great practical significance to choose the appropriate synthesis method for different application directions.

Here, we summarized the synthesis methods of ssDNA and divided the available approaches into the categories of chemical synthesis, enzyme synthesis, and bacteria-based synthesis. Chemical synthesis provides a powerful tool for the generation of custom ssDNA sequences. In fact, chemical synthesis is a method that can generate ssDNA without templates. Enzymatic synthesis encompasses the production of ssDNA by ligation or polymerization. Bacteria-based synthesis is a scalable approach that is based on fast-growing *Escherichia coli* (*E. coli*) cells as the host and offers mg-scale yields of ssDNA. This review focuses on representative methods of ssDNA synthesis in terms of both their principles and current challenges, providing a reference for researchers to better choose synthesis methods and thus increase the potential availability of ssDNA.

## 2. Chemical Synthesis

Single-stranded DNA fragments of less than 200 nt are mainly produced by direct chemical synthesis. The synthesis of short oligonucleotides is usually performed by different kinds of phosphoramidite chemistry methods, using either traditional column-based synthesizers or microarray-based synthesizers [29,30].

### 2.1. Column-Based Oligo Synthesis

The methodology of using phosphodiesters for the de-novo synthesis of oligos can be dated back to the 1950s, with pioneering studies that used phosphodiester, H-phosphonate, and phosphotriester approaches [31,32,33]. However, a true breakthrough in oligos synthesis did not occur until the early 1980s [34,35,36], with the development of solid-phase phosphoramidite approaches and automated instruments. This scalable oligonucleotide synthesis method offered sufficient robustness and fidelity and is used in the current commercial synthesis of oligonucleotides. Standard column-based oligo synthesis is a cyclical process that elongates a chain of nucleotides from the 3′-end to the 5′-end.

The synthesis proceeds on the base of standard phosphoramidite chemistry, which consists of a four-step chain elongation cycle (Figure 1) [3], encompassing (1) deprotection, (2) coupling, (3) capping (optional), and (4) oxidation [37]. The next synthesis cycle is then continued via the removal of the dimethoxytrityl (DMT) protecting group from the 5′-terminal. After the nucleoside sequence has been fully synthesized, the completed sequence is chemically cleaved from the solid support and the protecting groups are removed. Column-based synthesis is highly suitable for automated oligonucleotide synthesizers, which can produce 96 to 384 oligos simultaneously at scales from 10 to 100 nmol at costs ranging between $0.05 and $0.15 (in 2014 US dollars) per base [3]. Beyond this length, the efficiency of oligonucleotide production is generally reduced due to a drop in the yield and accumulation of small errors that are introduced in each step of the synthesis cycle [38]. Notably, the most recent work on improvements in the synthesis strategy and the utilized chemistries focused on increasing the length and quality of the synthesized oligonucleotides [29]. It is a universal approach that a number of different strategies can be used to introduce modifications in the oligos for different applications [39]. Through improvement of the chemistry synthetic steps, oligomers containing about 600 nucleotides should be synthesized in the future [40].

### 2.2. Array-Based Oligo Synthesis

With the introduction of microarray oligonucleotide synthesis by Affymetrix in 1990 [41,42], who developed photoactivation-based chemical methods for spatially located oligonucleotide synthesis, the foundation was laid for DNA microarrays. Array-based platforms offer a massively parallel alternative to traditional columnar oligonucleotide synthesis [43]. It is also a much cheaper strategy for oligo synthesis, with costs varying from $0.00001 to $0.001 per base (in 2014 US dollars) [3]. Furthermore, novel chip synthesis technology was developed to replace the chemical reactions on the surface of the light-guided array. For example, Agilent and Twist Bioscience developed an approach based on inkjet printing technology [38], which can synthesize 244,000 sequences 20 to 230 nts in length or 2000 to 696,000 sequences 120 to 300 nts in length in each pool [44]. Furthermore, the technology of CustomArray [45] can synthesize 12,472 (12k chip) or 92,918 (90k chip) sequences of 10 to 170 nts on a single chip [44]. For the application of DNA chips, it is necessary to design reasonably according to the purpose of the research [46]. In DNA information storage, using the random access approach, over 200 MB of data (35 distinct files) was encoded and stored in nine synthesis pools, which included 13,448,372 unique DNA sequences of lengths ranging from 150 to 154 bases [47]. In diagnostics, combining array-based oligo synthesis with photolithography computer chip technology, more than 400,000 oligonucleotides could be produced, which could detect up to 9000 genes on the glass surface of 1.6 square centimeters [48]. Although this platform offers superior synthesis capabilities and lower cost, there are still some challenges with using it for DNA synthesis applications [49]. The product yields of array-based oligo synthesis are typically at the femtomolar scale, i.e., two to four orders of magnitude lower than traditional column-based synthesis [29]. There have been many attempts to increase the scale, quality, and quantity of multi-channel synthesized oligonucleotides [50,51]. Continued improvements in array design, along with the optimization of synthesis reagents and processes, will hopefully deliver platforms for the synthesis of high-quality oligo sequences from arrays, potentially establishing them as the go-to source for multi-channel oligo sequence for gene synthesis applications.

## 3. Enzymatic Synthesis

Enzymatic synthesis is a low-cost, fast, and stable way to synthesize ssDNA. This process can directly synthesize the longer oligos due to its exquisite specificity and mild conditions. Efficient synthesis of ssDNA fragments ranging in size from several hundred base pairs to 10+ kb is needed for numerous biotechnology applications. However, methods for the construction of long ssDNA fragments of individual genes need to address a set of different challenges. Here, we focus on several methods that use enzymes to synthesize ssDNA (Figure 2 and Table 1).

### 3.1. Terminal Deoxynucleotidyl Transferase

Terminal deoxynucleotide transferase (TdT) is a polymerase that indiscriminately adds deoxynucleotide triphosphates (dNTPs) to the 3′ end of an ssDNA, which makes it a natural candidate for enzymatic ssDNA synthesis (Figure 2a) [62,63]. TdT is characterized by low substrate specificity for nucleotides and template-independent polymerization [64], which makes TdT-based ssDNA synthesis methods compatible with various modified nucleotides and convenient subsequent purification [65]. Recent studies have shown that the coupling time of C, G, and T is 1.5 min while that of A is 3 min [53]. The average step-yield is 97.7%, which is comparable with the performance of early phosphorylated amidine DNA synthesis [35,53]. A major challenge in TdT synthesis is the control of the addition of single bases, since TdT enzymes tend to catalyze the addition of multiple bases per cycle [65]. Although the synthesis of ssDNA by TdT is still an emerging strategy, several challenges remain. For example, it is still difficult to find a suitable solid support, there are problems with low extension yields, and the overall length is limited. Consequently, there is still no implementation of a practical enzymatic oligonucleotide synthesizer based on TdT, but recent application requirements indicate that this is a promising method. Potentially, TdT can be used to synthesize relatively long chains cheaply and quickly, and has been successfully applied in signal amplification [66], single-nucleotide modified in DNA oligos [65], polymerization of building blocks [67], and chain synthesis for DNA information storage [68].

### 3.2. Transcription and Reverse Transcription

In vitro transcription and reverse transcription (ivTRT) is a method that involves three steps: Preparation of dsDNA templates, transcription of RNA from the dsDNA, and preparation of ssDNA from the RNA (Figure 2b) [54]. Specifically, the dsDNA template is converted to an RNA via transcription, and then the RNA is reverted back to ssDNA using a reverse transcriptase. For in vitro transcription, PCR products or plasmids (containing a restriction site) can be used as the dsDNA templates, and transcription is performed using a T7 promoter and the very strong T7 RNA polymerase [69]. The RNA template is finally cleaved using RNase H, which leaves a heteroduplex structure at the 3′ end of the DNA [55]. The ivTRT can be used to synthesize ssDNAs of various lengths (about 0.5~2 kb), which can be used for gene editing [54,70,71]. Furthermore, when fluorescent-labeled deoxyuridine triphosphates (dUTPs) (Cy3-dUTP or Cy5-dUTP) existed in the pending-test RNA samples, the fluorescent probes (ssDNA) could be produced from single-round reverse transcription [48]. However, this method is both labor intensive and expensive. Moreover, the use of nucleases can limit the product yield and requires DNA of impeccable quality [71].

### 3.3. Asymmetric Polymerase Chain Reaction

Asymmetric polymerase chain reaction (aPCR) is the simplest method for effective production of ssDNA with on-demand labeling [72]. In theory, it is a straightforward ssDNA production protocol, and appeared after the publication of the PCR technique (Figure 2c) [73]. This method provides a way to direct the synthesis of ssDNA from a dsDNA template, and it has been used to generate ssDNAs ranging from hundreds to thousands of nucleotides [17]. In an aPCR reaction system, there are two amplification primers in unequal concentrations, and two phases of amplification are used to produce the desired ssDNA. The first one involves dsDNA templates’ exponential amplification, and the second one of linear amplification is used for producing ssDNA [74]. While this may seem simple, aPCR is prone to producing nonspecific amplification and therefore generally requires extensive experimentation to optimize the yield of the desired ssDNA [72]. Numerous research groups have attempted to improve aPCR by identifying the appropriate ratio of primers, polymerases, number of amplification cycles, and purification methods [56,75]. This approach has previously been applied in the synthesis of short ssDNAs [72,73,76], and more recently for DNA origami scaffolds of up to the kb scale [56,77]. High purity ssDNA can be produced by combining aPCR with gel purification or enzymatic degradation of residual chains. In fact, ssDNAs greater than 15 kilobases (kb) in length have been synthesized using aPCR, and a fluorescently modified ssDNA of 2000 nt was used to fold DNA nanoparticles [56]. In the systematic evaluation of ligand by exponential enrichments (SELEX) procedure aPCR, especially used for the amplification of short ssDNA libraries, but it is prone to the creation of by-products and nonseparated bands [78]. Several reports of different protocols indicate that the initial optimization of the aPCR is not easy, and the amplification of different ssDNA does not follow a unique pattern and hence cannot be done using a single protocol.

### 3.4. Isothermal Amplification of ssDNA

Isothermal amplification techniques for ssDNA production rely on enzyme activity or designed primers to bypass the thermal denaturation of the dsDNA template. The initiation step was shown to be the key factor that limits the speed and efficiency in the isothermal reaction. Consequently, initiation is also the main source of variance among the related methods [79]. Based on the excellent sensitivity, most isothermal amplification techniques were well established to detect DNA [80].

#### 3.4.1. Primer Exchange Reaction

A primer exchange reaction (PER) is a method that can isothermally produce ssDNA with custom DNA sequences in a programmable, autonomous, in situ, stepwise fashion with the aid of a strand-displacing polymerase (Figure 2d) [81]. The PER starts with the recognition and binding of a designed primer with an independent, customized sequence to its complementary sequence at the 3′-end of a catalytic hairpin structure. This hairpin contains a stop sequence that halts the polymerase-mediated extension reaction. The stop signal consists either of a G-C pair (if dGTP or dCTP are not used in the dNTP mix) or a modified base pair (i.e., iso-dG/iso-dC or methylated RNA) [57]. The newly extended primer then triggers the next round of extension in the programmable PER cascade that can autonomously synthesize DNA strand along the pre-prescribed pathway [57]. In the PER process, an arbitrary user-defined ssDNA strand is generated in situ only if the predesigned hairpin sets and the corresponding primer are both present. The PER cascade grows nascent strands of ssDNA with custom sequences for applications in nanodevices, signal amplifiers, logical computation, and molecular programming [58]. Although PER provides programmable synthesis of a user-specified ssDNA sequence in situ, the length is limited to 60 nt [82].

#### 3.4.2. Rolling Circle Amplification

Rolling circle amplification (RCA) relies on an isothermal polymerase (typically the large fragment of Bsu, Bst, and *E. coli* DNA Polymerase I, or Φ29 DNA polymerase) to synthesize a long stretch of repeating ssDNA sequences in a single unit (Figure 2e) [83]. An RCA reaction requires four factors: A DNA polymerase, a corresponding DNA primer, a template, and deoxynucleotide triphosphates (dNTPs) [84]. In RCA reaction, nucleotides are added continuously to the annealed primer by the polymerase, which generates a long ssDNA with a repeating sequence [85]. RCA is a powerful induction system because it can produce large amounts of ssDNA at the scale of a micron and a detectable amplification of a single molecule [84]. Examples for microgram-scale ssDNA production include using cutter hairpins [86] or annealing of a complementary digestion splint to form double-stranded restriction sites [87,88]. Due to its simplicity, robustness, and high sensitivity, RCA is considered a powerful tool for sensitive detection [89,90,91]. Nevertheless, there are some drawbacks to this approach. For example, the amplification efficiency is relatively low and the reaction time is as long as 6 h [92]. Variations of RCA for the production of ssDNA include linear RCA (LRCA), branched RCA (BRCA), hyperbranched RCA (HRCA), RCA with multiple primers (multi-primer RCA), and rolling-ring lock-type probe amplification [93].

#### 3.4.3. Other Isothermal Amplification Methods

The ssDNA produced by strand displacement amplification (SDA) or loop-mediated isothermal amplification (LAMP) has a high background, which makes separation a challenge [94]. However, the high sensitivity of these approaches has led to their wide use in other research areas [95,96,97,98]. Therefore, we offer a brief introduction to these methods. SDA is a nicking endonuclease-assisted isothermal polymerization reaction activated by four different specific primers, and its product is an ssDNA [98]. In a single SDA reaction, 10^9^ copies of the target DNA can be produced in less than an hour [99]. LAMP can amplify a few copies of DNA into a billion within an hour using the specially designed primer sets and a DNA polymerase [100,101]. The most vital step in the LAMP protocol is primer design. Usually, four to six primers are employed to specifically identify six to eight different regions of the target gene, and thereby amplify the gene with highly efficient precision [102]. The strand displacement activity is unique to the LAMP polymerase enzyme, which does not have an exonuclease activity at the 5′-3′, and therefore leads to the production of ssDNA [103].

### 3.5. Separation of ssDNA from dsDNA

Enzymatic or chemical approaches for the denaturation of dsDNA to form ssDNA offer an alternative strategy for ssDNA production but are often limited by the required purification steps [104]. There are many methods that can effectively generate ssDNA from dsDNA, usually relying on biotin–streptavidin separation [105], selective lambda-exonuclease digestion [106,107], denaturing urea polyacrylamide gel electrophoresis, and capillary zone electrophoresis [108].

For biotin–streptavidin-based separation of ssDNA (Figure 3a), one of the primers is biotinylated at the 5′ end, and the resulting biotinylated PCR product can be effectively fixed on to magnetic beads coated with streptomycin. Due to the high affinity between biotin and streptavidin, the desired non-biotinylated strand are separated from the biotinylated strands using denaturing treatment [109,110]. The resulting products are subsequently concentrated by ethanol precipitation or using a commercial purification kit. The separation efficiency of this method can reach 70% [111]. Although ssDNA separation via the biotin–streptavidin interaction is strongly favored, the biotinylated strand may reanneal with the desired strands, leading to an increased ratio of non-specific recovery [112].

Lambda exonuclease is an exodeoxyribonuclease that digests the phosphorylated strand from the 5′ to the 3′ end (Figure 3b), so that only non-phosphorylated ssDNA remains in the system after digestion [113]. Although this method is a fast and efficient method for generating ssDNA with high efficiency and quality [114], it has fallen out of favor due to the fact that incomplete digestion of the PCR product results in the accumulation of dsDNA in the final products [74].

Denaturing urea polyacrylamide gel electrophoresis (Figure 3c) can be used to visualize the isolated ssDNA. The differential migration of strands with different sizes on the urea-denaturing polyacrylamide (PAGE) gels can highly facilitate the selective recovery of desired strand. In the amplification step, one primer is modified with a modified group, such as ribose residues that are cleavable by ribonuclease, or a hexaethylene glycol tag, pH-labile base, or fluorophore, to separate two strands of different size, enabling ssDNA purification. This is a very efficient separation method because the resulting target ssDNA is clearly distinguishable. However, this method is not efficient in terms of labor and overall workflow since the electrophoresis on a denaturing gel followed by purification process of ssDNA takes a long time.

## 4. Bacteria-Based Production of ssDNA

Bacteria-based platforms for ssDNA synthesis offer milligram-scale yields in shake flasks, which can even be boosted further using bioreactors [115]. Bacteria-based ssDNA production employs bacteriophages with fast-growing *E. coli* cells as the host [115]. In the process of the secretion of progeny phage particles, the ssDNA genome is assembled into the virion with the coat proteins without lysing the host, so that these host cells can continue to divide after infection [116]. However, the fixed sequence of the M13 genome has limited application in ssDNA production [117]. Consequently, phagemids with the capacity of accommodating the size of several kb custom inserts were introduced to produce ssDNA. However, it typically includes a fixed region (2–3 kb) comprising a host origin of replication sequence, a phage origin from M13 or f1, and an antibiotic resistance gene, which limits their usefulness in nanotechnology [118,119]. At the DNA level (preparation of ssDNA, cloning, transfection efficiency), phagemid libraries are easier to work with than phages [120]. When phagemid-carrying cells are infected with the “helper phage” or transformed with a “helper plasmid”, an ssDNA with a near arbitrary sequence can be generated (Figure 4a) [119]. Although phagemid libraries can produce more and purer ssDNA than phage libraries, their application is complicated by the necessity of introducing an ssDNA replication origin and the selection of gene sequences, as well as their limitation to canonical deoxyribonucleotides [56]. The protocol has been used to produce various ssDNA scaffolds for the efficient assembly of DNA origami structures [121]. Notably, a λ/M13 hybrid virus was used to produce a circular ssDNA of 51,466 nts in *E. coli* [117]. Although phage-based single-stranded DNA production techniques are very mature, they still have several drawbacks. It is not easy to control the nature of the helper phage and the time of phage infection [118]. Furthermore, the use of a helper phage can be laborious, costly, and inefficient. It has also been shown that ssDNA can be produced in vivo using a reverse transcriptase protein [1]. However, the ssDNA must be incorporated into a long DNA with a complex secondary structure from which it would need to be cleaved in an additional step.

Retrons are distinct genomic DNA sequences found in many bacteria that code for a reverse transcriptase and a unique ssDNA/RNA hybrid, and have also been used for ssDNA production in vivo. Retrons consist of a single ~2000-bp operon containing a reverse transcriptase and two RNA moieties (msr and msd) (Figure 4b) [1,55]. The msr-msd cassette in the retron is folded into a secondary structure, then RT recognizes it and reverse transcribes the RNA sequence to produce a hybrid RNA-ssDNA molecule called multi-copy single-stranded DNA (msDNA) [122,123]. When co-expressed with the recombinase (RT protein, msr-msd RNA moieties, and β protein), the intracellularly expressed ssDNAs can introduce precise mutations into genomic DNA, thus transforming transient cellular signals into genome-encoded memories [1]. A reversion assay was used to measure the efficiency of DNA writing within living cells and the architecture of circuit where input, write, and read operations are independently controlled. In the related reports, ssDNAs of 32 to 205 nt were expressed using reverse transcriptases and either assembled into DNA nanostructures in vivo or purified for in vitro assembly, manufacturing, intracellular scaffolding, and imaging [124].

Interestingly, it was shown that the RC-replicating plasmid pC194 from Gram-positive bacteria can replicate and produce a circular-ssDNA in *E. coli* [125,126]. Thus, based on the replication mechanism of pC194, we constructed an engineering platform and demonstrated that ssDNA with various lengths and sequences can be produced in *E. coli* cells (own manuscript in revision).

## 5. Discussion and Conclusions

To enable precision engineering, as well as to broaden the application and design principles of synthetic biology techniques, recent innovations have sought to synthesize ssDNA efficiently. The methods of ssDNA synthesis, combined with accurate design and improvements in the quality of ssDNA, have made remarkable developments towards this goal. In this review, we summarized the representative methods used for ssDNA synthesis, including chemical, enzymatic, and bacteria-based approaches. We aimed at stimulating a broader interest in the continued development of efficient ssDNA synthesis techniques and improving their applications in synthetic biology, nanotechnology, and basic biological research. To date, different synthesis schemes were selected to obtain ssDNA for various applications. Hence, we hope to provide a bridge for researchers to better choose appropriate synthesis methods and enhance the availability of increasingly inexpensive synthetic ssDNA. As research into DNA synthesis techniques continues to progress, we anticipate technological innovations that are tailored for subsequent applications, promising to gradually reduce the synthesis time and cost.

## Figures and Tables

**Figure 1 genes-11-00116-f001:**
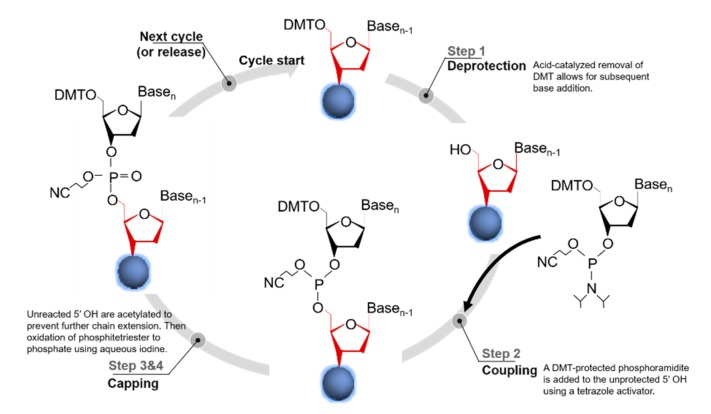
Phosphoramidite-based oligonucleotide synthesis [3].

**Figure 2 genes-11-00116-f002:**
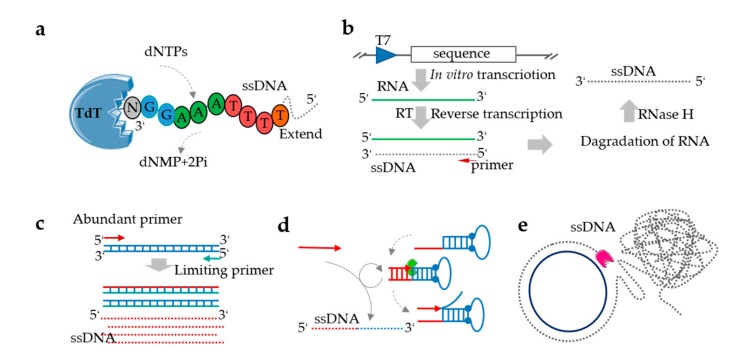
Schematic representation of enzymatic ssDNA synthesis. (**a**) The mechanism of terminal deoxynucleotide transferase (TdT)-based ssDNA synthesis; (**b**) The mechanism of transcription and reverse transcription; (**c**) The mechanism of asymmetric polymerase chain reaction (aPCR); (**d**) The primer exchange reaction (PER) cycle and mechanism for ssDNA synthesis; (**e**) The mechanism of rolling circle amplification (RCA).

**Figure 3 genes-11-00116-f003:**
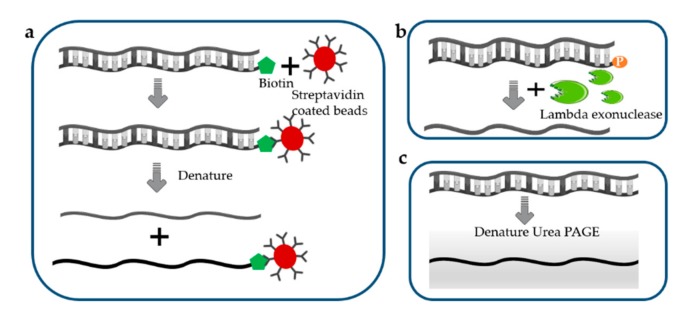
Direct separation of ssDNA. (**a**) Biotin–streptavidin separation of ssDNA; (**b**) Lambda exonuclease digestion; (**c**) Denaturing urea polyacrylamide gel electrophoresis.

**Figure 4 genes-11-00116-f004:**
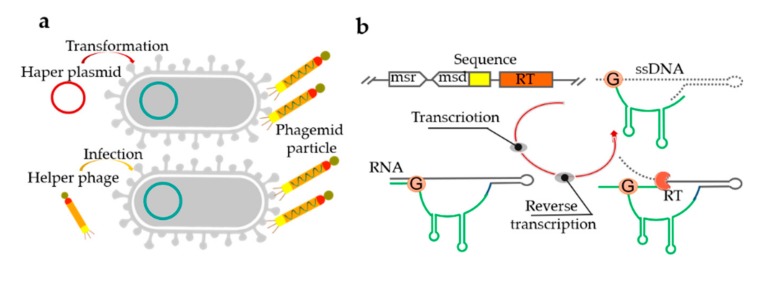
Bacteria-based ssDNA synthesis. (**a**) Schematic of the two approaches to phagemid-based ssDNA production. The phagemid-carrying *E. coli* cells are infected with the “helper phage” or transformed with a “helper plasmid”, the ssDNA can be generated. (**b**) The processes of ssDNA production by bacterial reverse transcriptases (RTs). In transcription step, the msr-msd RNA folds into a secondary structure; In reverse transcription step, the RT recognizes this secondary structure and uses a conserved guanosine residue as a priming site; Finally, a hybrid RNA-ssDNA molecule is produced.

**Table 1 genes-11-00116-t001:** Comparison of enzymatic methods for the synthesis of ssDNA.

Strategy	Template	Enzymes	Product Separation	Primer Design	Single Step Technique	Refs.
TdT	No	Terminal deoxynucleotide transferase	No	No	No	[52,53]
ivTRT	Yes	RNA polymerase, reverse transcriptase and RNaseH	Yes	Simple	No	[54,55]
aPCR	Yes	DNA polymerase	Yes	Simple	Yes	[56]
PER	Yes	DNA polymerase	Yes	Complex	Yes	[57,58]
RCA	Yes	DNA polymerase	Yes	Complex	Yes	[59,60]
SDA	Yes	DNA polymerase and strand-limited restriction endonuclease/nicking enzym	Yes	Complex	Yes	[61]

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
