# Peer review of "Current and Emerging Methods for the Synthesis of Single-Stranded DNA"

_genes, 2020, doi:10.3390/genes11020116_

Round 1
Reviewer 1 Report
To authors:
The authors review current methods to obtain ssDNA and they make the point of indicating aspects of each technique that need further development, thus stimulating interest in developing the method. This was the aim they stated in the abstract and in the discussion sections, and they did follow through.
The paper is interesting, but I found a number of things that would require corrections or would benefit from changes:
Line 38: The text refers to viral and bacterial genome reconstruction but the references deal with yeast. Although bacterial and viral genome reconstruction is mentioned in some of these papers, primary references to viral or bacterial genome reconstructions may be better.
Figure 2: I believe that the letters b) and c) are mixed in the figure. From the text, I think that where it says b) in the figure, it should be c) and vice versa.
Section 3.4.2.: I think that the last paragraph of this section may be better as a section of its own. That is, as section 3.4.3.
Figure 4. The caption for panel 4 should be extended with further explanation or, at the very least, state something like “see text for details”.
English: I noticed a number of grammatical errors. I list them below. Although the text is readable and understandable, a general review of the English usage and grammar would be convenient:
Line 44: I think it should be “for the synthesis of single-stranded DNA”
Line 78: I think it should be “The synthesis proceeds”
Line 83: I think it should be “suitable for automated oligonucleotide synthesizers”
Line 117: I think it should be “individual genes need to address”
Line 131: I think it should be “The average step-yield is 97,7%, which is comparable with”
Line 188: I think it should be “extended primer then triggers the next round of extension”
Line 191: I think it should be “The PER cascade grows nascent”
Lines 234 and 235: In the sentence “Although ssDNA separation via the biotin-streptavidin interaction is strongly favored but the biotinylated strand…” Either “although” or “but” should be removed.
Line 243: I would substitute “that” with a full stop and start a new sentence.
Line 301: I think it should be “to synthesize ssDNA efficiently”
Typos:
Line 141: The title “Transcription and reverse transcription” is missing its section number, which I believe should be 3.2., so that it would be “3.2. Transcription and reverse transcription”
Author Response
Dear reviewer,
Thank you for your letter regarding our manuscript entitled “Current and emerging methods for the synthesis of single-stranded DNA” that was submitted to genes for publication. We have carefully considered the critical comments and thoughtful suggestions, responded to these suggestions point-by-point, and revised the manuscript accordingly. We hope that our response is satisfactory, and we look forward to hearing from you soon. Please see the attachment.
With regards,
Hao Qi

Reviewer 2 Report
It is difficult to identify all the areas of application of single-stranded DNA. In general, they can be used as an object of investigation, an instrument for studying of natural structures and processes, and the agents of influencing on these processes. The manuscript (review) by Min Hao, Jianjun Qiao and Hao Qi is devoted current and emerging methods of the synthesis of single-stranded DNA. They summarized the representative methods used for ssDNA synthesis, including chemical, enzymatic and bacteria-based approaches and even technics used for isolation of single-stranded DNA from double-stranded ones. To say that summarizing this material is a difficult task is to say nothing. Not only reviews, but also books are devoted to each method. There are only few examples that may be mentioned in the text and in the list of References: Tatyana Abramova, Molecules, 2013; M. Caruthers, J. of Biological Chemistry, 2013; Methods in Molecular Biology, V.288, Oligonucleotide Synthesis; Nucleic Acids in Chemistry and Biology: Edition 3 or 4, Editors: G Michael Blackburn, Michael J Gait, David Loakes, David M Williams.
Since it is impossible to describe in detail all existing methods for obtaining single-stranded DNA in a short review, the authors should have focused in more details on the capabilities of each method. In the introduction and description of the chemical synthesis of oligonucleotides, there is no medical chemistry, that is, the creation of biomedical drugs and diagnostics. No information is available on the possibility of large-scale synthesis. Also, which approach allows you to introduce a modified residue in a given position of the oligonucleotide strand.
Small remarks:
Fig.2c – to change letters - in vitro translation
Line 141 – it must be 3.2
Lines 106 - 107 Sentence “Nevertheless…” requires continuation
Such a review may be useful to select a specific approach for obtaining single-stranded DNA for the upcoming study and based on available resources in the laboratory and may be published after adding some necessary details and correcting the mistakes.
Author Response
Dear reviewer,
Thank you for your letter regarding our manuscript entitled “Current and emerging methods for the synthesis of single-stranded DNA” that was submitted to genes for publication. We have carefully considered the critical comments and thoughtful suggestions, responded to these suggestions point-by-point, and revised the manuscript accordingly. We hope that our response is satisfactory, and we look forward to hearing from you soon.Please see the attachment
With regards,
Hao Qi
